# Diagnostic accuracy of S-Detect in distinguishing benign and malignant thyroid nodules: A meta-analysis

Lin Zhong[1], Cong Wang[2]*

**1** Pathology Department of the First Affiliated Hospital of Dalian Medical University, Dalian, China,
**2** Ultrasound Department of the First Affiliated Hospital of Dalian Medical University, Dalian, China

* wc027214@163.com

## Abstract

### Objectives

In this meta-analysis study, the main objective was to determine the accuracy of S-detect in effectively distinguishing malignant thyroid nodules from benign thyroid nodules.

### Methods

We searched the PubMed, Cochrane Library, and CBM databases from inception to August 1, 2021. Meta-analysis was conducted using STATA version 14.0 and Meta-Disc version 1.4 softwares. We calculated summary statistics for sensitivity (Sen), specificity (Spe), positive and negative likelihood ratio ($LR^+/LR^-$), diagnostic odds ratio(DOR), and receiver operating characteristic (SROC) curves. Cochran's Q-statistic and $I^2$ test were used to evaluate potential heterogeneity between studies. A sensitivity analysis was performed to evaluate the influence of single studies on the overall estimate. We also performed meta-regression analyses to investigate the potential sources of heterogeneity.

### Results

In this study, a total of 17 studies meeting the requirements of the standard were used. The number of benign and malignant nodules analyzed and evaluated in this paper was 1595 and 1118 respectively. This paper mainly completes the required histological confirmation through s-detect. The pooled Sen and pooled Spe were 0.87 and 0.74, respectively, (95%CI = 0.84–0.89) and (95%CI = 0.66–0.81). Furthermore, the pooled $LR^+$ and negative $LR^-$ were determined to be 3.37 (95%CI = 2.53–4.50) and 0.18 (95%CI = 0.15–0.21), respectively. The experimental results showed that the pooled DOR of thyroid nodules was 18.83 (95% CI = 13.21–26.84). In addition, area under SROC curve was determined to be 0.89 (SE = 0.0124). It should be pointed out that there is no evidence of bias (i.e. t = 0.25, P = 0.80).

**Data Availability Statement:** All relevant data are within the paper and Supporting Information files.

**Funding:** The author(s) received no specific funding for this work.

**Competing interests:** The authors have declared that no competing interests exist.

## Conclusions

Through this meta-analysis, it can be seen that the accuracy of s-detect is relatively high for the effective distinction between malignant thyroid nodules and benign thyroid nodules.

## Introduction

In the past period, the incidence rate of thyroid related diseases is increasing, which is closely related to the progress in biological characteristics and ultrasound diagnosis technology [1]. For the clinical diagnosis of this disease, ultrasonic examination method is usually selected, and its main advantage is that it has very high sensitivity [2]. In recent years, various new ultrasonic imaging technologies, such as ultrasound elastography, contrast-enhanced ultrasound, and superb micro-vascular imagine, have developed rapidly, which has greatly promoted their application in the field of medicine [3–5].

However, its accuracy is usually affected by the professional ability of doctors. Computer-aided diagnosis (CAD) technology is a popular topic in artificial intelligence and modern medical research [6]. Ultrasonic S-Detect technology is typically a new diagnosis method that uses a convolutional neural network deep learning algorithm to evaluate thyroid nodules according to the Ti-RADS dictionary, at present, it is the most widely used CAD diagnosis system in this field. The deep learning model is used to automatically detect and analyze the boundary, shape, internal echo, and other nodule information, overcome the interference of human factors, and objectively judge benign and malignant nodules [7]. A large number of research data show that if this technology is applied to the clinical diagnosis of thyroid nodules, it will obtain good accuracy [8–10]. The problem to be pointed out is that the application time of this technology is still relatively short, so there are some different views. In addition, there are some differences in the clinical results. For example, the Spe obtained by Yoo et al. is 88% [9], but the value obtained by other team is 41%, which is quite different [11].

According to the existing data, it is found that there are significant differences in the reported Spe. At present, there is no meta-analysis study on its application in the diagnosis of thyroid cancer. Therefore, this paper will focus on in-depth analysis and research in this aspect.

## Methods

This study was conducted in accordance with the PRISMA (Preferred Reporting Items for Systematic Reviews and MetaAnalyses) guidelines, the meta-analysis was not registered.

### Literature search

We searched PubMed, Cochrane Library, and CBM databases from inception to August 1, 2021. The following keywords and MeSH terms were used: ["thyroid cancer" or "thyroid neoplasm" or "thyroid tumor" or "thyroid nodule "] and ["S-Detect" or "smart detect" or "artificial Intelligence" or "computer-aided diagnosis"]. We also performed a manual search to identify additional potential articles.

### Selection criteria

The following four criteria were required for each study: (1) the study design must be a clinical cohort study or diagnostic test; (2) the study must relate to the accuracy of S-Detect for the

differential diagnosis of benign and malignant thyroid nodules,and the final assessments from S-Detect were in dichotomized form: possibly benign and possibly malignant; (3) all thyroid nodules were histologically confirmed; and(4) published data in the fourfold (2×2) tables must be sufficient. If the study did not meet all the inclusion criteria, it was excluded. The most recent publication or publication with the largest sample size was included when the authors had published several studies using the same subjects.

## Data extraction

Relevant data were systematically extracted from all included studies by two researchers using a standardized form. The researchers collected the following data: first author's surname, publication year, language of publication, study design, sample size, number of lesions, source of the subjects, "gold standard". True positives (TP), true negatives (TN), false positives (FP), and false negatives (FN) in the fourfold (2 × 2) tables were also collected.

## Quality assessment

Methodological quality was independently assessed by two researchers using the Quality Assessment of Studies of Diagnostic Accuracy Studies (QUADAS) tool [12]. The QUADAS criteria include 14 assessment items. Each item was scored as "yes" (2), "no" (0), or "unclear"(1). The QUADAS score ranged from 0 to 28, and a score $\geq$ 22 indicated good quality.

## Statistical analysis

STATA version 14.0 (Stata Corp, College Station, TX, USA) and Meta-Disc version 1.4 (Universidad Complutense, Madrid, Spain) software were used for meta-analysis. We calculated the pooled summary statistics for sensitivity (Sen), specificity (Spe), positive and negative likelihood ratio (LR+/LR−), and diagnostic odds ratio (DOR) with 95%confidence intervals (CIs). The post-test probabilities were calculated by the $LR^+$and $LR^-$ and plotted on a Fagan nomogram. A summary receiver operating characteristic (SROC) curve and corresponding area under the curve (AUC) were obtained. The threshold effect was assessed using Spearman's correlation coefficients. Cochran's Q-statistic and $I^2$ test were used to evaluate potential heterogeneity between studies. If significant heterogeneity was detected(Q test P<0.05, or I test>50%), a random-effects model or fixed-effects model was used. We also performed meta-regression analyses to investigate the potential sources of heterogeneity. A sensitivity analysis was performed to evaluate the influence of single studies on the overall estimate. We used Begger's funnel plots and Egger's linear regression tests to investigate publication bias.

## Results

### Characteristics of included studies

Initially, the search keywords were used to identify 40 articles. We reviewed the contents of the title and abstract of the article, and then listed 17 of its value. On this basis, we further reviewed the integrity of the content and data of the paper, and then excluded 6 papers. In the current meta-analysis, a total of 17 studies were used [7–10, 13–25]. In Fig 1 of this paper, the selection method and main steps of the article are described in detail. The numbers of benign and malignant thyroid nodules analyzed and evaluated in this paper were 1595 and 1118 respectively. In S1 Table, the research methods and basic characteristics adopted are shown in detail. We found that QUADAS scores were $\geq$ 24.

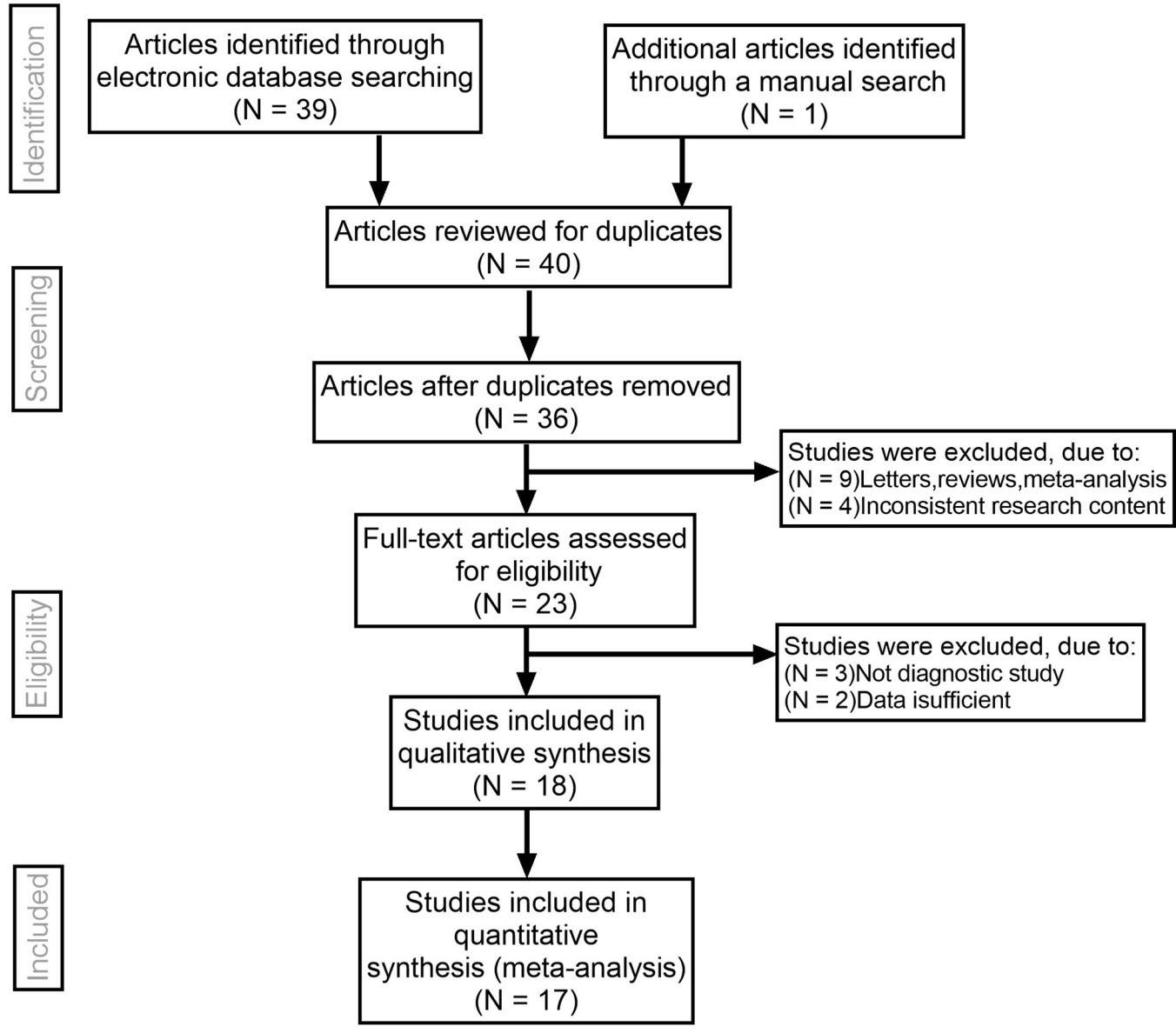

**Fig 1. The research selection and literature retrieval process of this paper (a total of 17 studies).**

### Quantitative data synthesis

We chose the random effect model, the main reason is that there is no very significant difference between different studies. We also conducted sensitivity analysis and found that the results were not affected (Fig 2). It is found that the pooled Sen and Spe were 0.87 (95% CI = 0.84–0.89) and 0.74 (95%CI = 0.66–0.81) respectively (Fig 3). The experimental results showed that there was no obvious correlation between the specificity and the sensitivity (r = 0.289, P = 0.260), which means that no threshold effect exists. It can be seen that the negative LR- and pooled LR+ were 0.18 (95%CI = 0.15–0.21) and 3.37 (95%CI = 2.53–4.50) respectively (Fig 4). The study found that after using S-Detect for this diagnosis, the DOR value was 18.83 (95% CI = 13.21–26.84) (Fig 5). In addition, the area below the SROC curve was equal to 0.89 (SE = 0.0124) (Fig 6). Meta-regression analysis results confirmed that no factor could

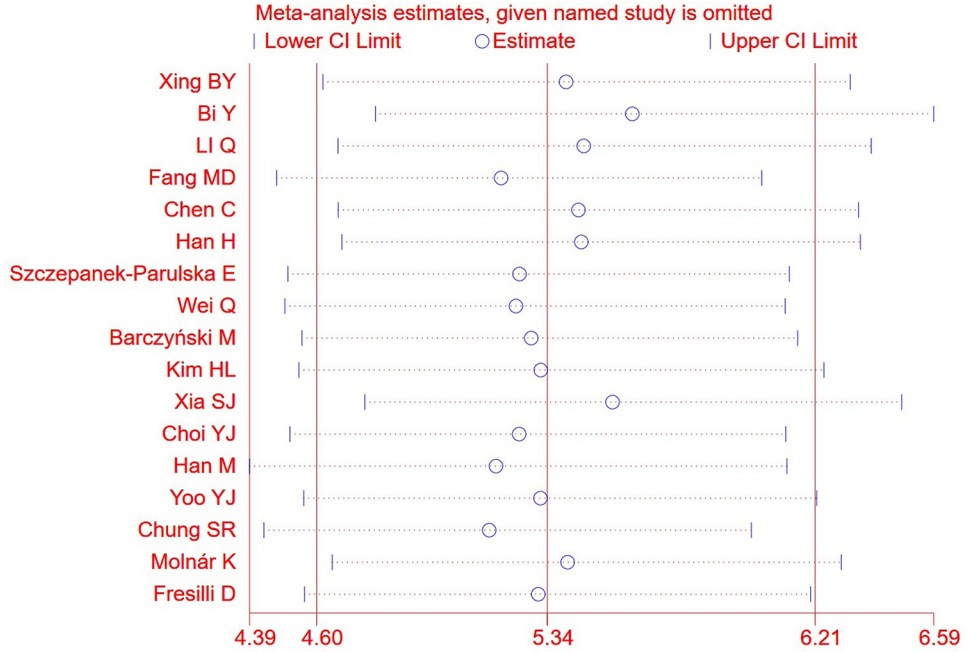

**Fig 2. The sensitivity analysis carried out in this paper.** They did not significantly affect the results.

explain the potential sources of heterogeneity (S2 Table). Through analysis, no evidence of publication bias was observed (in Fig 7). After egger's test, there was also no evidence of publication bias (t = 0.25, P = 0.80). According to the research on Fagan diagram, when the pre-test

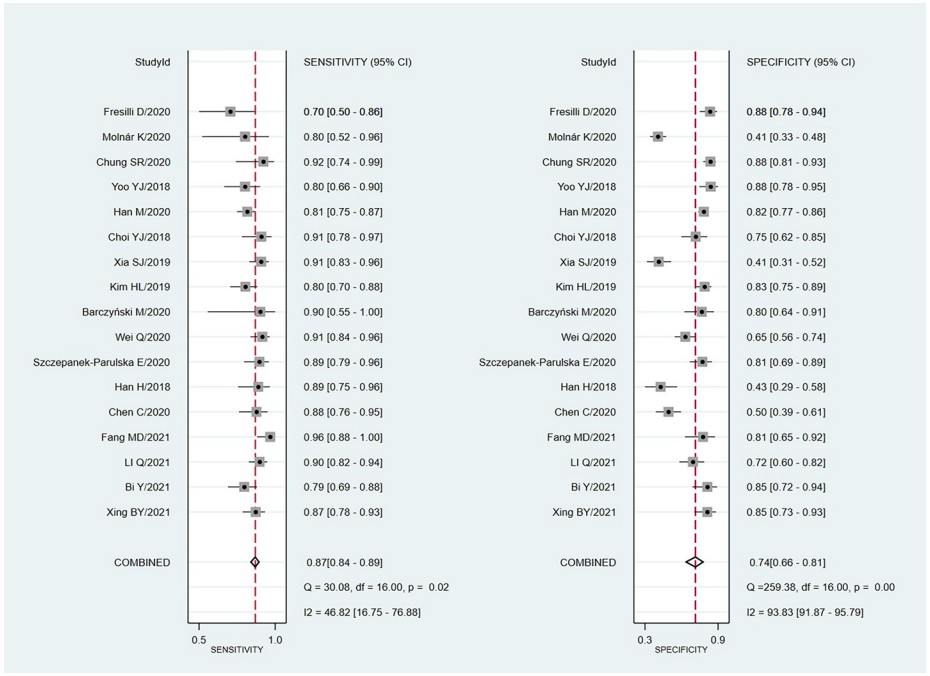

**Fig 3. The Forest map results of specificity and sensitivity of S-Detect for diagnosis.**

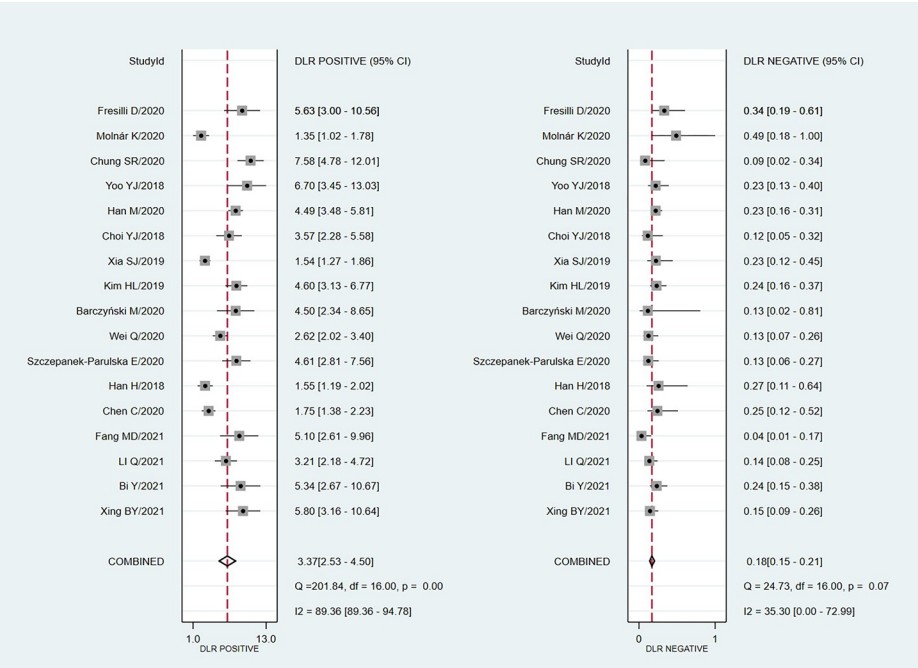

**Fig 4. The forest diagram results of the likelihood ratio between negative and positive of S-Detect diagnosis obtained in this paper.**

probabilities were 25%, 50% and 75%, the negative and positive post-test probabilities were 6%, 15% and 35%, as well as 53%, 77% and 91%, respectively (Fig 8).

## Discussion

As a very common disease, the study of thyroid nodule has attracted extensive attention. The related identification will affect the final clinical decision. At present, high-resolution ultrasound technology has been widely used in this field [25]. In recent years, the incidence rate of US has increased with the increasing incidence of the disease. In fact, this imaging method has obvious performance advantages. With the rapid development of artificial intelligence (AI) technology, it also drives the continuous improvement of this field. For example, "S-Detect" technology is a typical representative. This technology is helpful to better complete the morphological analysis of the disease, so as to promote the formulation of its clinical treatment plan [13–18]. However, only a few articles have reported its diagnostic performance for thyroid cancer, and they were mainly published by Korean researchers. To further study the diagnostic value of S-detect in thyroid ultrasound, more validation sets from different countries are required. Therefore, Our main purpose is to promote the better application of S-Detect in the field of clinical diagnosis of thyroid tumors.

This paper mainly analyzes the main performance and effect of s-detect in the analysis and diagnosis of thyroid nodules. We studied 17 subjects and evaluated 1595 benign and 1118 malignant nodules, respectively. The data obtained show that its comprehensive Sen value was 0.87, Spe value was 0.74 and DOR value was 18.83. The above results show that S-Detect has high accuracy in the field of clinical diagnosis of thyroid nodules, so it is a very good diagnostic tool. In general, we believe that threshold effect is a significant change that often occurs after a phenomenon has exceeded the relevant range. According to our analysis, we can find that

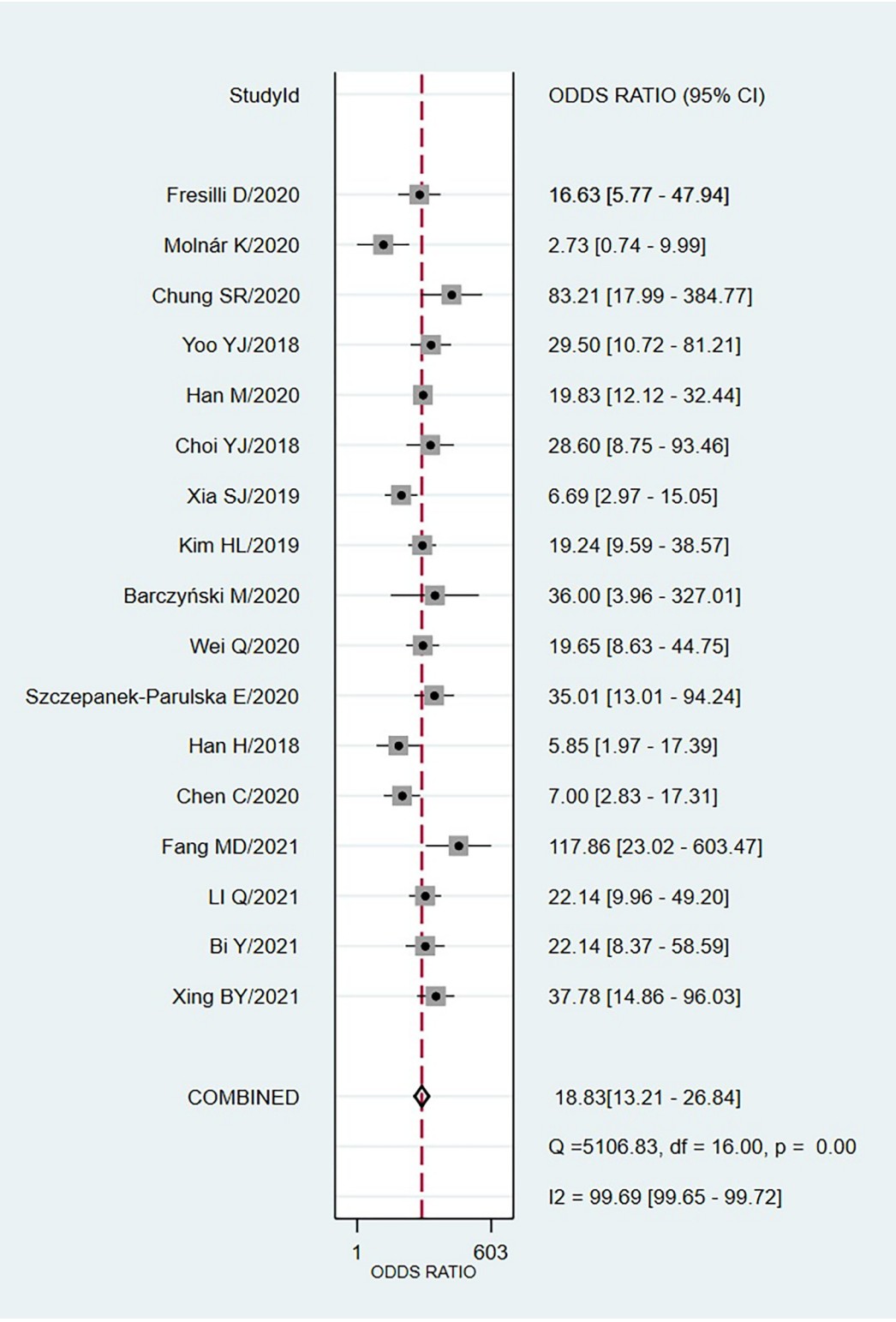

**Fig 5. The forest plot results of DOR of S-Detect for diagnosis.**

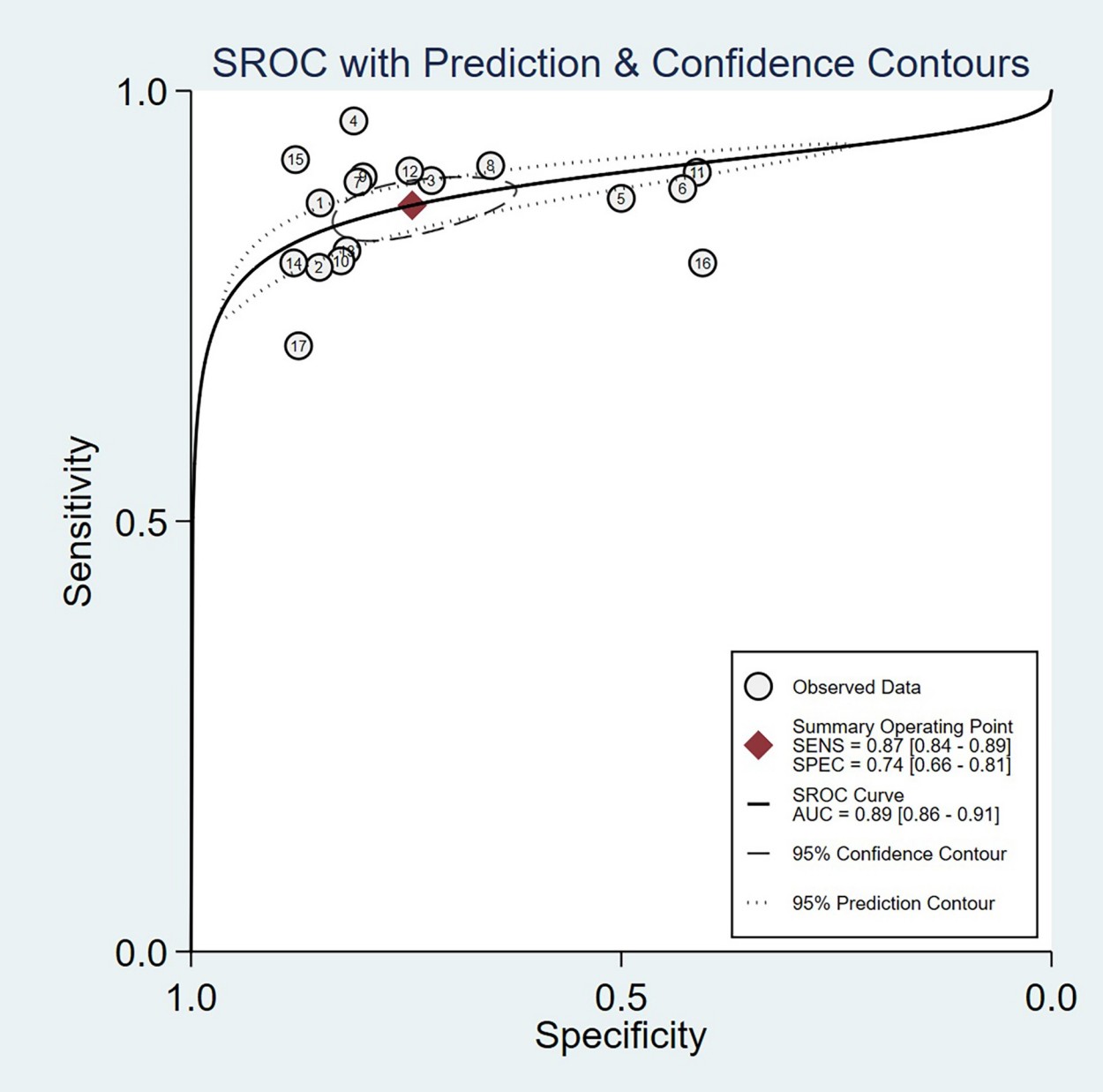

**Fig 6. SROC curve of S-Detect diagnostic accuracy obtained in this paper.**

there is no very obvious internal relationship between Sen and Spe, that is, there is no sufficient evidence related to the threshold effect. In the current study, we found no evidence of publication bias. Therefore, the current research confirms that S-Detect has good accuracy, and the result is consistent with the original report.

S-Detect has good accuracy in the field of clinical diagnosis of thyroid nodules, but there are still some limitations in this study. Firstly, the quality of the sample needs to be further improved, because its accuracy needs to be further improved. Secondly, the inclusion of studies with histological confirmation only and the retrospective nature of the meta-analysis could have led to subject selection bias. Thirdly, we did not find the information about registration

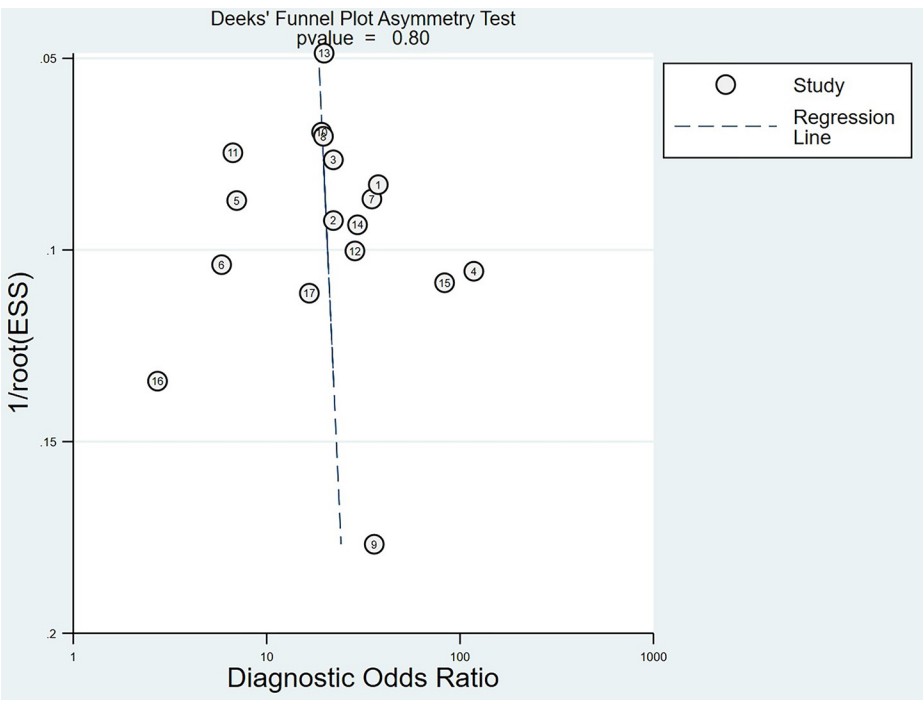

**Fig 7. The results of Begger's funnel plot of publication bias on the pooled OR.** There is no publication bias observed.

and version of the used S-Detect software, which may be a potential source of heterogeneity. In particular, it should be pointed out that the current research objects are basically from Asian countries, which will affect the effectiveness of the current results.

According to the results of this meta-analysis, S-Detect can accurately distinguish malignant thyroid nodules from benign thyroid nodules. It is also an important auxiliary means of conventional ultrasonic examination technology. Because the research process is limited by a series of factors, we need to explore this problem in depth.

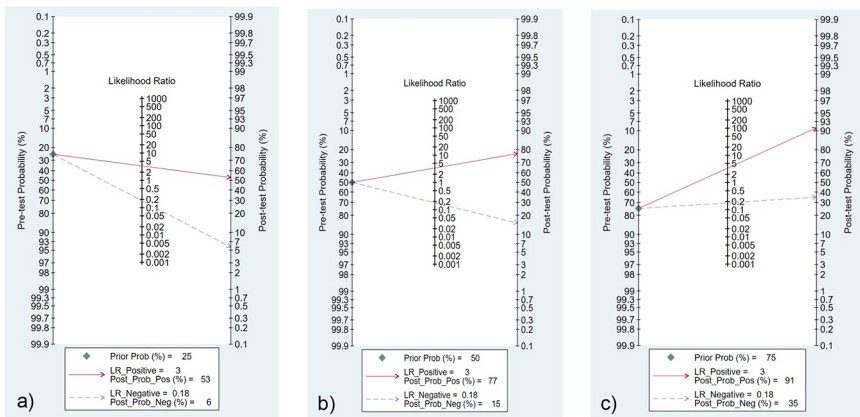

**Fig 8.** The Fagan diagram analysis results obtained in this paper: a)—c) pre inspection probability is 25%, 50% and 75% respectively.

## Supporting information

**S1 Checklist. PRISMA checklist.**
(DOC)

**S1 Table. Baseline characteristics and methodological quality of all included studies.**
(DOCX)

**S2 Table. Meta-regression analyses of potential source of heterogeneity.**
(DOCX)

**S1 File. Search strategy.**
(DOCX)

**S1 Data.**
(XLS)

## Author Contributions

**Conceptualization:** Lin Zhong.

**Formal analysis:** Cong Wang.

**Investigation:** Lin Zhong.

**Methodology:** Cong Wang.

**Resources:** Lin Zhong.

**Software:** Cong Wang.

**Visualization:** Cong Wang.

**Writing – original draft:** Lin Zhong.

**Writing – review & editing:** Cong Wang.

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
