## [Decision Letter · Decision Letter 0]

5 Apr 2022

PONE-D-21-26569

Diagnostic accuracy of S-Detect in distinguishing benign and malignant thyroid nodules: a meta-analysis

PLOS ONE

Dear Dr. Wang,

Thank you for submitting your manuscript to PLOS ONE. After careful consideration, we feel that it has merit but does not fully meet PLOS ONE’s publication criteria as it currently stands. Therefore, we invite you to submit a revised version of the manuscript that addresses the points raised during the review process.

We look forward to receiving your revised manuscript.

Kind regards,

Nguyen Quoc Khanh Le

Academic Editor

PLOS ONE

Journal Requirements:

2. Thank you for stating the following in your Competing Interests section: "None"

5. We noticed you have some minor occurrence of overlapping text with the following previous publication(s), which needs to be addressed:

- https://medultrason.ro/medultrason/index.php/medultrason/article/view/2460

- https://www.researchsquare.com/article/rs-16146/v1

In your revision ensure you cite all your sources (including your own works), and quote or rephrase any duplicated text outside the methods section. Further consideration is dependent on these concerns being addressed.

Reviewers' comments:

Reviewer's Responses to Questions

**Comments to the Author**

1. Is the manuscript technically sound, and do the data support the conclusions?

Reviewer #1: Partly

Reviewer #2: Partly

2. Has the statistical analysis been performed appropriately and rigorously? 

Reviewer #1: Yes

Reviewer #2: Yes

3. Have the authors made all data underlying the findings in their manuscript fully available?

Reviewer #1: Yes

Reviewer #2: Yes

4. Is the manuscript presented in an intelligible fashion and written in standard English?

Reviewer #1: No

Reviewer #2: No

5. Review Comments to the Author

Reviewer #1: The authors report a meta-analysis to evaluate the diagnostic accuracy of S-Detect in distinguishing benign and malignant thyroid nodules.

Introduction is quite general, and some statements are misleading. For example, "the sample sizes were not enough" (lines 56-57) should be better explained and referenced.

Keywords. The following keywords and MeSH terms were used: ["thyroid cancer" or "thyroid neoplasm" or "thyroid tumor" or "thyroid nodule "] and [“S-Detect” or “smart detect” or “artificial Intelligence” or "computer aid diagnosis” or “machine intelligent”] (lines 60-62). Some keywords seem to be confusing. What is the meaning of "machine intelligent"? It returns only ten results in Pubmed when directly used as a keyword.

Data extraction (lines 75-78). The researchers collected the following data: the first author's surname, publication year, language of publication, study design, sample size, number of lesions, source of the subjects, "gold standard," and diagnostic accuracy.

I think the researchers collected the number of true positives, true negatives, false positives, and false negatives, instead of diagnostic accuracy. Is this correct?

Furthermore, was the S-Detect version used registered? At least 2 different versions exist.

Moreover, how was the index test interpreted? S-DetectTM for Thyroid is a technology providing the features of the selected nodule with information on 6 features. Instead of directly suggesting malignancy or benignity of the nodules, the software may classify them according to three major thyroid imaging and data reporting systems: K-TIRADS, Russ, and ATA guidelines. So, did the authors collect the mode in which S-Detect was used in each study and the threshold used to "diagnose" malignant cases? Please, clarify.

I think these are important issues that should be added to the meta-regression analysis.

Why was the country of origin of the study not used as a potential variable in the meta-regression analysis?

Reference standard and Selection bias. The inclusion of studies with histological confirmation only is a strength (because histology is a good reference standard), but also a limitation (because surgical series are prone to selection bias). The authors should briefly discuss this issue.

Registration. Was this meta-analysis pre-registered in a public database? (e.g., PROSPERO). Please, clarify.

There are some typos and grammar errors that should be corrected.

Reviewer #2: This meta-analysis aimed to identify the diagnostic accuracy of S-Detect 15 in distinguishing benign and malignant thyroid nodules. I have several suggestions.

1. Please provided the search strategy in each database.

2. Please provide the Fagan diagram.

3. The heterogeneity within studies are very high, and please performed the subgroup analysis or meta-regression.

4. The PLR, NLR and DOR should be reported, too.

5 Please evaluate the threshold effect withing study.

6. PLOS authors have the option to publish the peer review history of their article (what does this mean?). If published, this will include your full peer review and any attached files.

Reviewer #1: No

Reviewer #2: No

---

## [Decision Letter · Decision Letter 1]

24 Jun 2022

PONE-D-21-26569R1Diagnostic accuracy of S-Detect in distinguishing benign and malignant thyroid nodules: a meta-analysisPLOS ONE

Dear Dr. Wang,

Thank you for submitting your manuscript to PLOS ONE. After careful consideration, we feel that it has merit but does not fully meet PLOS ONE’s publication criteria as it currently stands. Therefore, we invite you to submit a revised version of the manuscript that addresses the points raised during the review process.

We look forward to receiving your revised manuscript.

Kind regards,

Nguyen Quoc Khanh Le

Academic Editor

PLOS ONE

Journal Requirements:

Reviewers' comments:

Reviewer's Responses to Questions

**Comments to the Author**

1. If the authors have adequately addressed your comments raised in a previous round of review and you feel that this manuscript is now acceptable for publication, you may indicate that here to bypass the “Comments to the Author” section, enter your conflict of interest statement in the “Confidential to Editor” section, and submit your "Accept" recommendation.

Reviewer #1: (No Response)

Reviewer #2: All comments have been addressed

2. Is the manuscript technically sound, and do the data support the conclusions?

Reviewer #1: Partly

Reviewer #2: Yes

3. Has the statistical analysis been performed appropriately and rigorously? 

Reviewer #1: Yes

Reviewer #2: Yes

4. Have the authors made all data underlying the findings in their manuscript fully available?

Reviewer #1: Yes

Reviewer #2: Yes

5. Is the manuscript presented in an intelligible fashion and written in standard English?

Reviewer #1: Yes

Reviewer #2: Yes

6. Review Comments to the Author

Reviewer #1: The authors addressed my previous questions.

However, some minor edits are still needed:

1. Search strategy. I highlighted an issue with the search criteria. While I agree that the specific keyword is not modifying the query results, the point is: are the authors sure that their strategy missed no appropriate paper? I see that the other reviewer also asked for clarification and complete queries.

The new query (“computer aid diagnosis”) is misspelled (computer-aided diagnosis being the correct spelling).

2. The authors did not find the information about registration and version of the used S-Detect software. It should be discussed as a limitation of the meta-analysis.

3. The country where the study was performed should be included as a variable in the meta-regression analysis reported in Table 2, if possible. If not possible, it should at least be discussed as a limitation.

Reviewer #2: The author had addressed all comments. I have not other questions, and this study can be accepted for publicatoin.

7. PLOS authors have the option to publish the peer review history of their article (what does this mean?). If published, this will include your full peer review and any attached files.

Reviewer #1: No

Reviewer #2: No

---

## [Editor Report · Decision Letter 2]

14 Jul 2022

Diagnostic accuracy of S-Detect in distinguishing benign and malignant thyroid nodules: a meta-analysis

PONE-D-21-26569R2

Dear Dr. Wang,

We’re pleased to inform you that your manuscript has been judged scientifically suitable for publication and will be formally accepted for publication once it meets all outstanding technical requirements.

Kind regards,

Nguyen Quoc Khanh Le

Academic Editor

PLOS ONE
---

## [Editor Report · Acceptance letter]

27 Jul 2022

PONE-D-21-26569R2 

Diagnostic accuracy of S-Detect in distinguishing benign and malignant thyroid nodules: a meta-analysis 

Dear Dr. Wang:

I'm pleased to inform you that your manuscript has been deemed suitable for publication in PLOS ONE. Congratulations! Your manuscript is now with our production department. 

Kind regards, 

on behalf of

Dr. Nguyen Quoc Khanh Le 

Academic Editor

PLOS ONE